Review Article

# Signaling roles for astrocytic lipid metabolism in brain function

Juan P Bolaños [ID] [1,2,3][✉] & Angeles Almeida [ID] [1,2][✉]

## Abstract

**Astrocytes, the most abundant glial cell type in the central nervous system, have traditionally been viewed from the perspective of metabolic support, particularly supplying neurons with lactate *via* glycolysis. This view has focused heavily on glucose metabolism as the primary mode of sustaining neuronal function. However, recent research challenges this paradigm by positioning astrocytes as dynamic metabolic hubs that actively engage in lipid metabolism, especially mitochondrial fatty acid β-oxidation. Far from serving solely as an energy source, fatty acid ß-oxidation in astrocytes orchestrates reactive oxygen species-mediated signaling pathways that modulate neuron-glia communication and cognitive outcomes. This review integrates recent advances on astrocytic fatty acid ß-oxidation and ketogenesis, alongside other metabolic pathways converging on reactive oxygen species dynamics, including cholesterol metabolism and peroxisomal β-oxidation. In reframing astrocytic metabolism from energy provision to signaling, we propose new directions for understanding central nervous system function and dysfunction.**

**Keywords** Astrocytes; Fatty Acid β-Oxidation; Ketogenesis; Reactive Oxygen Species Signaling; Neuron-glia Metabolic Coupling
**Subject Categories** Metabolism; Neuroscience; Signal Transduction

## Introduction

Astrocytes are pivotal regulators of central nervous system (CNS) homeostasis, with critical roles spanning neurotransmitter clearance, ion buffering, synapse modulation, and metabolic support. Their unique positioning, ensheathing blood vessels and synapses, places them at the nexus of systemic circulation and neuronal networks. This anatomical advantage allows astrocytes to serve as intermediaries, translating peripheral metabolic states into CNS responses (Bonvento and Bolanos, 2021; Hosli et al, 2022; Kacem et al, 1998). Despite this strategic role, astrocyte metabolism has historically been predominantly characterized by glycolysis and the astrocyte–neuron lactate shuttle (Almeida et al, 2004; Barros et al, 2005; Magistretti and Allaman, 2015; Zimmer et al, 2017).

The classical glycolytic model posits that astrocytes metabolize glucose to produce lactate, which is then shuttled to neurons for oxidative metabolism. This concept, while instrumental in shaping our understanding of brain energy metabolism (Bolaños and Magistretti, 2025) restricts astrocyte function to a supportive role focused mostly on energy supply. However, emerging evidence suggests that astrocytes exhibit significant metabolic flexibility, adapting their substrate utilization based on local and systemic cues. Transcriptomic and metabolomic studies reveal robust lipid catabolism in astrocytes (Eraso-Pichot et al, 2018; Fecher et al, 2019; Morant-Ferrando et al, 2023), challenging the notion of astrocytes as strictly glycolytic. This lipid metabolism, particularly mitochondrial fatty acid ß-oxidation (FAO), is not merely an alternative energy source but serves as a crucial regulator of mitochondrial function, reactive oxygen species (ROS) signaling, and neuronal activity (Ioannou et al, 2019; Morant-Ferrando et al, 2023). Moreover, lactate derived from astrocytic glycolysis should not be conceptualized solely as a metabolic substrate: it also functions as a signaling molecule. In neurons, extracellular L-lactate can bind to the G protein-coupled receptor Hcar1 (also known as Gpr81), triggering downstream signaling (e.g., via Gi, lowering cAMP) that can modulate neuronal excitability and synaptic activity (Lauritzen et al, 2014). Thus, beyond fueling oxidative metabolism, astrocyte-derived lactate may act in a paracrine and, according to recent data, autocrine (Fernandez-Moncada et al, 2024), fashion to influence neuronal signaling dynamics. This dual role energetic support plus receptor-mediated modulation- suggests a more integrative view of astrocyte–neuron metabolic coupling, in which the lactate shuttle is also a vector for metabolic signaling, potentially coordinating local energy supply with neuronal responsiveness.

In this review, we explore the multifaceted roles of astrocytic FAO, emphasizing its signaling functions over its energetic contributions. We integrate FAO within broader metabolic contexts, including ketogenesis, cholesterol metabolism, peroxisomal β-oxidation, and the pentose-phosphate pathway (PPP), all converging on ROS dynamics. We further discuss astrocytes' role as metabolic sensors of peripheral status, shaping CNS responses through these interconnected pathways. By broadening the lens on astrocyte metabolism, we propose a shift from viewing metabolism

[1]Institute of Functional Biology and Genomics (IBFG), University of Salamanca, CSIC, Salamanca, Spain. [2]Institute of Biomedical Research of Salamanca (IBSAL), University Hospital of Salamanca, Salamanca, Spain. [3]Centro de Investigación Biomédica en Red de Fragilidad y Envejecimiento Saludable (CIBERFES), Madrid, Spain.
[✉]E-mail: jbolanos@usal.es; aaparra@usal.es

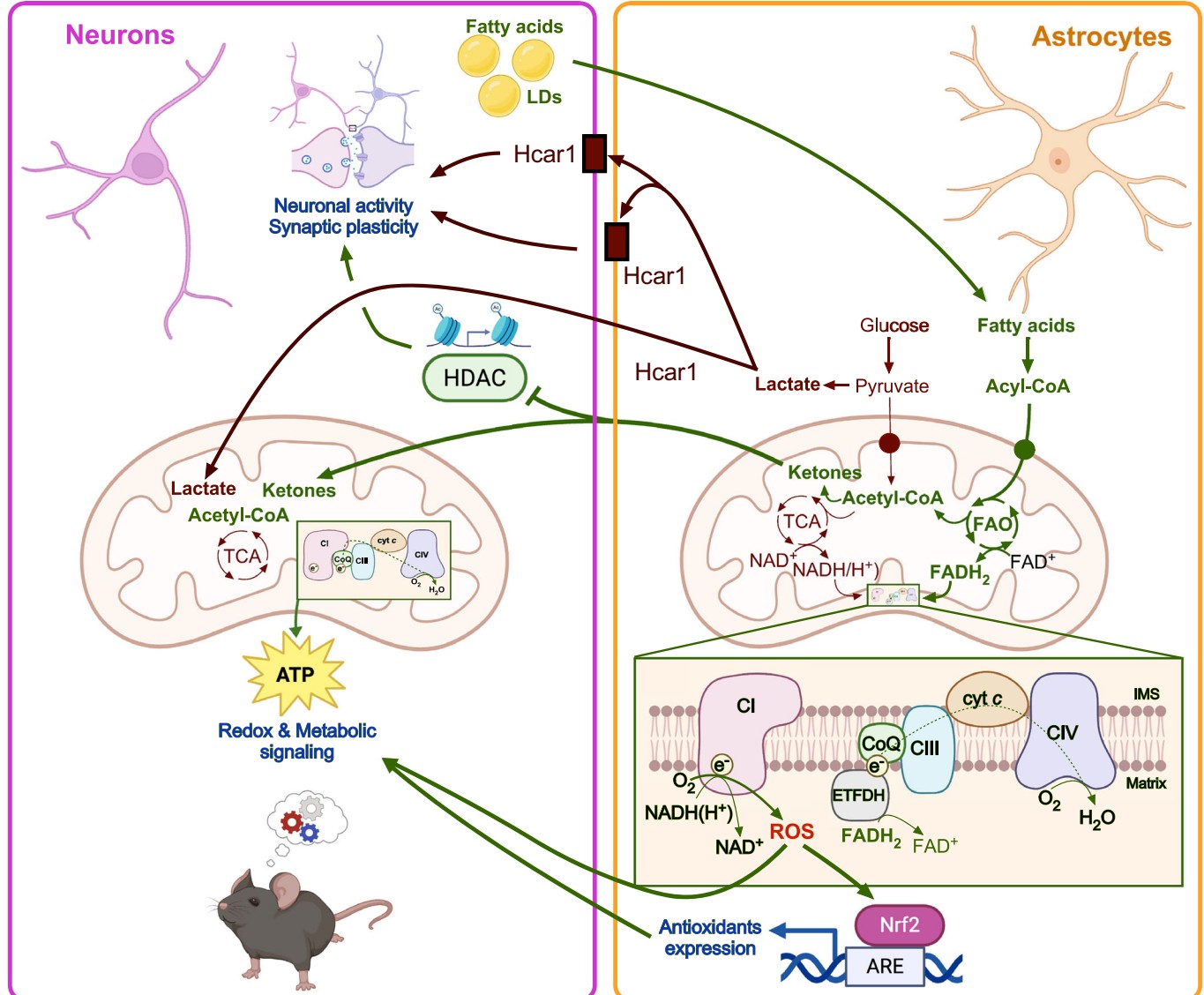

**Figure 1. Astrocytic fatty acid ß-oxidation and signaling.**

In astrocytes, fatty acid ß-oxidation (FAO) sustains FADH$_2$ regeneration, which transfers electrons into the electron transport chain directly to ubiquinone (CoQ), bypassing mitochondrial complex I (CI), weakening the interaction between CI and complex III (CIII), a configuration that favors reactive oxygen species (ROS) formation by free CI. This physiological mitochondrial ROS generation targets specific metabolic and redox substrates, including nuclear factor erythroid 2-related factor 2 (Nrf2), which transcriptionally activates the antioxidant response elements (ARE) to boost an antioxidant response that contributes to maintain the redox status of neighbor neurons. Peroxidated fatty acids from neuronal lipid droplets (LDs) can be shuttled to astrocytes to sustain FAO. This metabolic configuration in astrocytes favors acetyl-CoA conversion into ketones, which may be shuttled to neurons, where they can modulate chromatin by inhibiting histone deacetylase (HDAC), thus linking astrocytic metabolism to epigenetic regulation in neurons. Ketones, as lactate—which is also a signaling metabolite *via* the Hcar1 receptors—can also be used as a metabolic fuel to sustain energy generation. Thus, astrocytic FAO plays a metabolic and signaling processes that sustain synaptic and neuronal function. Created with BioRender.

as a mere energy supply chain to recognizing its central role in CNS signaling and function.

## Astrocytic fatty acid β-oxidation: beyond ATP production

Fatty acid β-oxidation is a catabolic process in which long-chain fatty acids undergo sequential degradation within mitochondria, producing acetyl-Coenzyme A (CoA), FADH$_2$, and NADH(H$^+$).

While FAO is well-known for its role in energy production in peripheral tissues like muscle and liver, its function in astrocytes is unique. FAO-derived FADH$_2$ reduces ubiquinone *via* the electron-transferring-flavoprotein dehydrogenase (ETFDH) (Guarás et al, 2016). Compared with neurons, the contribution of ß-oxidation via ETFDH to the reduction of ubiquinone in the electron transport chain (ETC) is higher in astrocytes, leading to a weaker interaction between complex I (CI) and complex III (CIII) (Morant-Ferrando et al, 2023). Consequently, astrocytic mitochondria exhibit a greater proportion of free CI, which favors ROS generation through CI-

mediated electron leakage (Lopez-Fabuel et al, 2016) (Fig. 1). Interestingly, in a recent report, it has been shown that, in addition to this basal, physiological CI-mediated ROS formation (Lopez-Fabuel et al, 2016), astrocytic CIII contributes to mitochondrial ROS generation upon exposure to neuropathological stimuli promoting dementia (Barnett et al, 2025). However, in contrast to the conventional view of ROS as harmful byproducts, astrocytic CI-mediated ROS act as secondary messengers, modulating redox-sensitive pathways and neuron-glia communication. Notably, ROS derived from FAO in astrocytes are essential for memory consolidation and cognitive function, as demonstrated both in rodent (Vicente-Gutierrez et al, 2019; Morant-Ferrando et al, 2023) and *Drosophila* models (Rabah et al, 2025). Interfering with astrocytic FAO disrupts mitochondrial ROS signaling and impairs synaptic plasticity, underscoring the central role of this metabolic pathway in brain function.

Noteworthy, neurons are also capable of FAO (Kumar et al, 2025). Although historically considered metabolically constrained in this regard, recent genetic and functional studies reveal that neurons can engage mitochondrial β-oxidation under specific conditions. The enzyme $Ddhd_2$ (phospholipase A1) facilitates the release of saturated fatty acids (myristate, palmitate, stearate) within neurons, which then undergo mitochondrial β-oxidation to support neuronal ATP production during peak demand (Greda et al, 2025; Saber et al, 2025). Yet, evidence from transcriptomics and functional assays indicates that astrocytes are overall much more competent in FAO than neurons. In particular, astrocytes express higher levels of key FAO genes (e.g., *Cpt1a*, *Acadl*, *Mtpα*) relative to neurons, and their mitochondrial respiration is more sensitive to the Cpt1 inhibitor etomoxir (Morant-Ferrando et al, 2023). In fact, astrocytes take up peroxidized lipids released from neurons and metabolize them through ß-oxidation (Ioannou et al, 2019). These data support a model where neuronal FAO is likely subordinate, whereas astrocytes act as the principal site of fatty acid catabolism in the brain. This asymmetry allows astrocytes not only to buffer lipid substrates but also to modulate local redox dynamics and ROS signaling, as well as metabolic coupling with neurons (Morant-Ferrando et al, 2023) (Fig. 1).

## Astrocytic ketogenesis: fueling neurons and modulating signaling

While FAO in astrocytes produces acetyl-CoA, this metabolite is predominantly channeled into ketogenesis rather than fueling the TCA cycle (Morant-Ferrando et al, 2023). Ketogenesis in the liver is a well-characterized response to low-glucose states, producing ketone bodies -β-hydroxybutyrate (BHB), acetoacetate (AcAc), and acetone- as alternative fuels for peripheral tissues. Also, astrocytes possess the enzymatic machinery for local ketone body synthesis (Blazquez et al, 1998; Blazquez et al, 1999; Guzman and Blazquez, 2001), positioning them as an auxiliary source of ketones within the CNS. Whilst cultured neurons can perform FAO (Greda et al, 2025; Saber et al, 2025), there is no in vivo demonstration that these cells efficiently perform ketogenesis—i.e., FAO-derived acetyl-CoA conversion into ketone bodies. This evolutionarily conserved capacity of astrocytes to perform FAO (Schulz et al, 2015; McMullen et al, 2023) would allow them to locally support neurons with ketone bodies (Silva et al, 2022; Guzman and Blazquez, 2001;

McMullen et al, 2023; Schulz et al, 2015; Silva et al, 2022), at least in *Drosophila*. Notably, beyond their role as energy substrates, ketone bodies exert significant signaling functions. BHB, for example, acts as an inhibitor of histone deacetylases (HDACs), modulating gene expression and promoting neuroprotective pathways (Newman and Verdin, 2014; Puchalska and Crawford, 2017). Thus, ketogenesis would contribute to neuron-glia communication, possibly not only by supplying fuel, but also by influencing transcriptional programs within neurons.

Moreover, ketone bodies modulate redox balance and mitochondrial function. BHB enhances antioxidant defenses by upregulating nuclear factor erythroid 2-related factor 2 (Nrf2)-mediated transcription and can reduce oxidative stress within neurons (Kolb et al, 2021). These signaling properties further reinforce the importance of astrocytic ketogenesis as a neuroprotective mechanism, extending its function beyond metabolic support. The local generation of ketone bodies by astrocytes therefore represents an integrated metabolic-signaling axis by modulating cellular pathways that govern plasticity and survival.

Emerging evidence suggests that astrocytic ketogenesis also interacts with systemic metabolic states. During fasting or ketogenic diets, peripheral ketone levels rise for the use by the brain—particularly neurons—as oxidative fuels to sustain energy for neurotransmission under a condition in which glucose availability is scarce (Garcia-Rodriguez and Gimenez-Cassina, 2021). However, astrocytes may also supplement this with locally produced ketones, hence contributing to neuronal function. This dual-source model of ketogenesis underscores the importance of astrocytic lipid metabolism in responding to metabolic fluctuations and maintaining neuronal function (Monda et al, 2024). These multifaceted roles position astrocytic ketogenesis as a critical contributor to CNS resilience, particularly under metabolic stress. Yet, the full extent of its signaling roles in health and disease remains to be elucidated (Won et al, 2025). Future research should explore how astrocytic ketogenesis integrates with other metabolic pathways and signaling networks within the brain, particularly in the context of neurodegenerative diseases, when energy metabolism is often disrupted.

## ROS signaling: the nexus between astrocytic metabolism and neuronal function

Reactive oxygen species are classically regarded as detrimental byproducts of mitochondrial oxidative metabolism, implicated in aging and neurodegeneration. However, recent studies have redefined ROS as crucial signaling entities that orchestrate cellular communication, redox balance, and plasticity both in neurons and in astrocytes (Doser et al, 2024; Oswald et al, 2018; Sies et al, 2022; Vicente-Gutierrez et al, 2019). As indicated above, this signaling role of ROS, at least in astrocytes, emerges prominently from the unique configuration of their mitochondria, where FAO-derived $FADH_2$ fuels electron flux into the ETC, leading to disassembled CI states that favor physiological ROS generation (Lopez-Fabuel et al, 2016; Morant-Ferrando et al, 2023).

The ROS produced under these conditions, primarily superoxide and its downstream product hydrogen peroxide ($H_2O_2$), act locally within astrocytes to modulate transcription factors such as Nrf2, which regulates the expression of antioxidant enzymes including glutathione

peroxidase and superoxide dismutase (Vicente-Gutierrez et al, 2019). This antioxidant response is essential for maintaining the redox environment in astrocytes and for protecting neurons from oxidative stress. Furthermore, beyond astrocytic self-regulation, ROS serve as intercellular messengers. Astrocyte-derived $H_2O_2$ can diffuse to adjacent neurons, where it influences synaptic function and plasticity (Vicente-Gutierrez et al, 2019). Moreover, astrocytic ROS modulate neuronal activity and behavior. Reducing ROS abundance by expressing a mitochondrial form of catalase (Vicente-Gutierrez et al, 2019), or by inhibiting FAO *via* knocking out *Cpt1a* (Morant-Ferrando et al, 2023), selectively in astrocytes in the adult mouse, impairs memory formation. Moreover, in *Drosophila* models, it has been shown that an astrocytic-to-neuronal $H_2O_2$ signaling supported long-term memory formation, and was impaired in Alzheimer's disease models (Rabah et al, 2025).

Importantly, ROS signaling from astrocytes integrates with broader redox networks within the brain. For example, in mouse, astrocytic mitochondrial ROS downmodulate extracellular NADPH(H$^+$) oxidases, keeping extracellular ROS reduced, influencing glia-neuron signaling dynamics (Vicente-Gutierrez et al, 2019). This species-conserved, layered ROS signaling framework (Rabah et al, 2025; Vicente-Gutierrez et al, 2019) underscores its complexity and necessity for proper brain function. Astrocyte-derived ROS are therefore not mere metabolic byproducts but critical messengers that coordinate metabolic and synaptic networks. Their generation, regulated by FAO and ETC dynamics, provides a mechanistic link between astrocyte metabolism and higher-order brain functions like cognition and memory. Future directions should focus on elucidating the specific redox-sensitive signaling cascades in neurons triggered by astrocyte-derived ROS and how these are dysregulated in neurodegenerative conditions.

## Additional lipid metabolic pathways converging on signaling

While mitochondrial FAO and ketogenesis constitute central lipid metabolic processes in astrocytes, other lipid pathways also contribute critically to CNS homeostasis and signaling. One such pathway is cholesterol metabolism. Astrocytes are the principal source of cholesterol in the adult brain, synthesizing and supplying this essential lipid to neurons *via* apolipoprotein E (ApoE)-containing lipoproteins (Pfrieger and Ungerer, 2011; Vanherle et al, 2025). Cholesterol is vital for synapse formation, myelin maintenance, and membrane fluidity. The transfer of cholesterol from astrocytes to neurons directly impacts neuronal excitability and synaptic plasticity (Mauch et al, 2001). Notably, disruptions in cholesterol metabolism, particularly involving the ApoE4 isoform, have been implicated in Alzheimer's disease pathogenesis, highlighting the intersection between lipid metabolism and neurodegeneration (Qi et al, 2021; Vanherle et al, 2025).

Beyond its structural roles, cholesterol metabolism intersects with signaling pathways. Oxysterols, derivatives of cholesterol metabolism, act as ligands for nuclear receptors such as liver X receptors (LXRs), which regulate genes involved in lipid homeostasis, inflammation, and immunity (Vanherle et al, 2025). In astrocytes, Lxr activation modulates inflammatory responses and promotes cholesterol efflux, maintaining a balanced lipid environment that supports neuronal health (Vanherle et al, 2025).

Dysregulation of oxysterol signaling can thus exacerbate neuroinflammation and impair neuron-glia communication.

Another critical lipid pathway is peroxisomal β-oxidation, which complements mitochondrial FAO by degrading very-long-chain fatty acids (VLCFAs) and branched-chain fatty acids (Vanherle et al, 2025). This degradation prevents the accumulation of toxic lipid species that could disrupt membrane integrity and signaling. Unlike mitochondrial FAO, peroxisomal β-oxidation produces hydrogen peroxide ($H_2O_2$) as a byproduct, contributing to cellular redox balance and signaling (Ding et al, 2021). Ding *et al* demonstrated that peroxisomal β-oxidation regulates lipolysis in astrocytes via $H_2O_2$-mediated signaling, linking lipid degradation with metabolic adaptation.

Sphingolipid metabolism contributes to astrocyte signaling through both intracellular pathways and intercellular communication. Intracellularly, sphingolipid intermediates such as ceramides and sphingosine-1-phosphate (S1P) modulate astrocyte apoptosis, redox regulation, and inflammatory gene expression (Gault et al, 2010; Hannun and Obeid, 2008). In parallel, astrocytes release bioactive sphingolipids -particularly S1P- that act on neighboring neurons and glial cells through G protein-coupled receptors to influence synaptic plasticity and neuroinflammatory tone (Zhang et al, 2023). Thus, astrocytes not only maintain sphingolipid homeostasis within their own metabolic networks but also use sphingolipid derivatives as cell-cell signaling cues. Dysregulation of these processes is linked to pathological neuroinflammation and neurodegeneration, as observed in multiple sclerosis and Alzheimer's disease (Zhang et al, 2023). Integrating these sphingolipid-mediated signals with mitochondrial FAO and ketogenesis highlights astrocyte lipid metabolism as a regulator of CNS communication beyond its classical energetic functions (Fig. 2).

## Astrocytes as peripheral metabolic sensors: integrating systemic cues into CNS signaling

Astrocytes, strategically positioned at the interface between blood vessels and neurons, are ideally suited to function as metabolic sensors, integrating systemic signals into CNS responses. Their endfeet directly contact the blood-brain barrier (BBB), enabling them to detect fluctuations in circulating metabolites, hormones, and inflammatory mediators (Abbott et al, 2006; Hosli et al, 2022; Kacem et al, 1998). Through this anatomical and functional connection, astrocytes transduce peripheral metabolic states, such as nutrient availability, hormonal changes, and systemic inflammation, into local adaptations that affect neuronal function and behavior.

Astrocytes express receptors for several key metabolic hormones, including insulin, leptin, and glucagon-like peptide-1 (Glp-1) (Fuente-Martin et al, 2012; Garcia-Caceres et al, 2016; Timper et al, 2020). These receptors enable astrocytes to respond dynamically to systemic energy status. For example, Glp-1 receptor signaling in astrocytes modulates FAO, mitochondrial integrity, and inflammatory responses, thereby linking peripheral nutrient sensing with astrocytic metabolic pathways and brain homeostasis (Timper et al, 2020). Similarly, insulin signaling in astrocytes regulates glucose uptake and glycogen storage, influencing their support for neurons during energy fluctuations (Garcia-Caceres et al, 2016). Disruption of these signaling pathways can exacerbate CNS pathology, particularly in metabolic disorders such as obesity

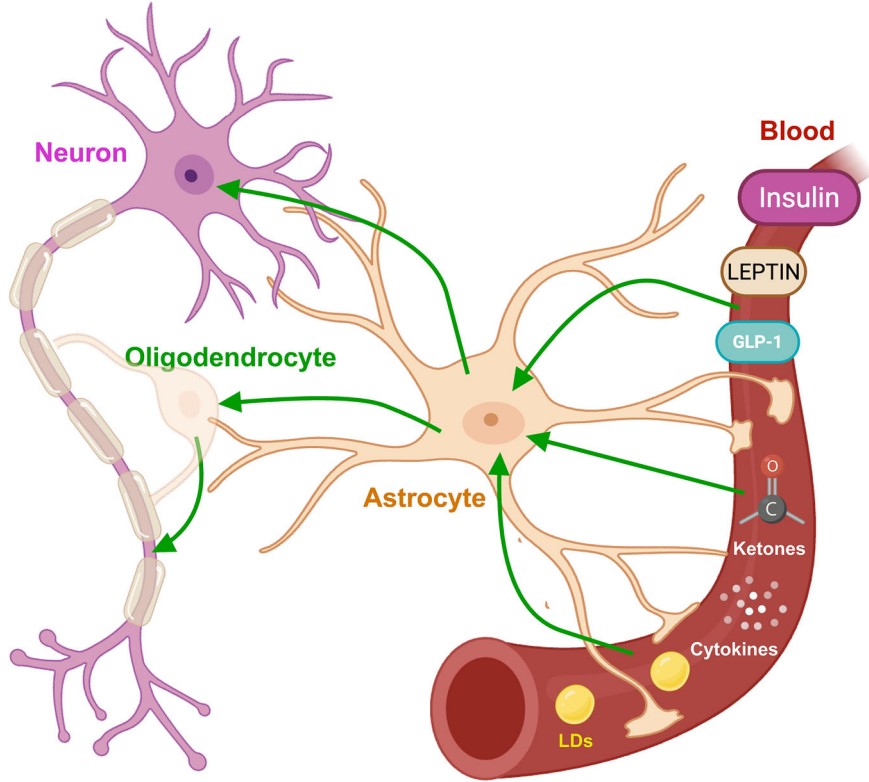

**Figure 2. Astrocytic fatty acid metabolism as an integrated signaling hub.**

Astrocytic endfeet contact blood vessels and detect circulating hormones and metabolites, including insulin, leptin, glucagon-like peptide-1 (GLP-1), ketone bodies, and inflammatory cytokines, as well as lipid droplets (LDs) and fatty acids. In response, astrocytes adapt their lipid metabolism and fatty acid β-oxidation, modulating reactive oxygen species signaling and producing ketone bodies. These signals are relayed to neighboring neurons and oligodendrocytes, thereby influencing synaptic activity, myelin integrity and overall brain homeostasis. Created with BioRender.

and diabetes, which are associated with increased neuroinflammation and cognitive decline.

Recent studies suggest that astrocytes also sense peroxidized lipids from neuronal origin or peripheral lipids and fatty acids, adjusting their metabolic programs accordingly (Ioannou et al, 2019; Liu et al, 2017; Vanherle et al, 2025). In conditions of high circulating fatty acids, such as obesity or fasting, astrocytes adapt by increasing FAO and ketogenesis, modulating ROS levels and neuroinflammation (Vanherle et al, 2025). In addition to hormonal and nutrient sensing, astrocytes detect peripheral inflammatory signals. They express pattern recognition receptors (Prrs), including toll-like receptors (Tlrs), which respond to systemic inflammatory mediators (Phulwani et al, 2008). This positions astrocytes as intermediaries between systemic inflammation and CNS immune responses, further integrating peripheral signals into brain homeostasis. The chronic activation of these pathways in metabolic syndromes or infections can exacerbate neuroinflammatory responses, impacting synaptic function and neuronal survival (Heneka et al, 2015).

Thus, astrocytes function as critical nodes in the communication between the periphery and the brain, integrating hormonal, nutrient, lipid, and inflammatory signals. These interactions enable astrocytes to modulate their own metabolic pathways, such as FAO, ketogenesis, and lipid metabolism, and thereby influence neuronal function and behavior. Understanding this astrocyte-

mediated integration is essential for unraveling the complex interplay between systemic metabolism and CNS function, especially in the context of metabolic and neurodegenerative diseases.

## Lipid metabolism in oligodendrocytes: supporting axonal integrity and neuronal function in white matter

While this review emphasizes astrocytic lipid metabolism as a central regulator of neuronal function via redox signaling and ketogenesis, it is important to recognize the distinct yet complementary role of lipid metabolism in other glial cell types, particularly oligodendrocytes. Oligodendrocytes are the primary myelinating cells of the CNS, forming the myelin sheath that insulates axons and facilitates rapid electrical signal conduction. Myelin is a lipid-rich structure, and oligodendrocytes exhibit specialized lipid metabolic processes to maintain its integrity and function.

Recent studies have demonstrated that oligodendrocytes engage in mitochondrial β-oxidation of fatty acids, not only to meet their own energetic demands but also to support axonal function and neuronal health in white matter tracts (Asadollahi et al, 2024; Nave and Werner, 2014; Saab and Nave, 2017). This metabolic activity is

particularly critical during remyelination and in response to axonal injury, where lipid remodeling and energy provision are essential for repair and regeneration. Interestingly, lipid metabolism in oligodendrocytes also contributes to intercellular metabolic coupling with axons. Endurance exercise, such as marathon running, causes a substantial reduction in myelin water fraction—a proxy of myelin content—in specific brain regions involved in motor coordination and sensory and emotional integration, but recovers within 2 months (Ramos-Cabrer et al, 2025). This study suggests that systemic physiological challenges like prolonged exercise may upregulate lipid metabolic pathways in oligodendrocytes to sustain axonal energy demands, although this interpretation is still speculative and would require direct demonstration. If so, it would reinforce the emerging concept that oligodendrocytic lipid metabolism dynamically responds to peripheral metabolic states. In fact, oligodendrocytes can transfer metabolites, including lactate and lipids, to axons, supporting their energy metabolism under stress conditions (Funfschilling et al, 2012; Nave et al, 2023; Saab et al, 2013; Saab et al, 2016). This metabolic support is particularly vital in long white matter tracts, where axons are distant from their neuronal cell bodies and local energy provision is crucial (Fig. 2).

**Box 1  In need of answers**

1. How is astrocytic fatty acid β-oxidation dynamically regulated by neuronal activity and systemic metabolic states in vivo?
2. To which defined neuronal targets do FAO-derived ROS from astrocytes shape synaptic plasticity and cognition?
3. How are lipid metabolic tasks partitioned between astrocytes and oligodendrocytes to support axonal integrity and myelin dynamics?
4. Can selective modulation of astrocytic FAO or ketogenesis be exploited therapeutically to ameliorate neurodegenerative diseases?

## Conclusion and future directions

Astrocytes are no longer mere supporters of neuronal function through metabolic provisioning; they are dynamic regulators of CNS signaling, leveraging lipid metabolism, particularly FAO, ketogenesis, cholesterol metabolism, peroxisomal β-oxidation, and sphingolipid pathways, to orchestrate redox signaling and neuron-glia communication. These processes converge on ROS dynamics, positioning astrocytes at the heart of metabolic and signaling networks that sustain cognitive function and CNS homeostasis. Future research (Box 1) should prioritize elucidating the crosstalk between these metabolic pathways, particularly their interactions with systemic metabolic cues and redox signaling. Further exploration of how astrocytic lipid metabolism influences neuroinflammation, synaptic plasticity, and neurodegeneration will enhance our understanding of brain metabolism beyond its classical energetic framework. Investigating therapeutic strategies that modulate astrocytic metabolism, such as targeting FAO, ketogenesis, or cholesterol transport, may offer novel interventions for metabolic and neurodegenerative diseases. In embracing this expanded perspective, the field can move beyond traditional energy-centric views of brain metabolism, recognizing the signaling

roles of astrocytic lipid pathways in maintaining CNS function and resilience.

## Peer review information

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

## Acknowledgements

JPB is funded by MICIU/AEI (PID2022-138813OB-I00 /10.13039/501100011033 and FEDER, UE), la Caixa Foundation (grant agreement LCF/PR/HR23/52430016), the European Union's Horizon Europe research and innovation program under the MSCA Doctoral Networks 2021 (101072759; FuEl ThEbRaiN In healtThY aging and age-related diseases, ETERNITY, and the European Research Council (ERC) Advanced Grant NeuroSTARS (ref. 101199747). A.A. is funded by Instituto de Salud Carlos III (PI21/00727, RD21/0006/0005, PMP22/00084; Junta de Castilla y León (CSI011P23 and Escalera de Excelencia CLU-2017-03) and MICIU/AEI (RED2022-134407-T).

## Author contributions

**Juan P Bolanos**: Conceptualization; Funding acquisition; Writing—original draft; Writing—review and editing. **Angeles Almeida**: Conceptualization; Funding acquisition; Writing—original draft; Writing—review and editing.

## Disclosure and competing interests statement

The authors declare no competing interests.

