## [Peer Review File · EMBO Reports]

Signaling roles for astrocytic lipid metabolism in brain function

Juan Bolanos and Angeles Almeida

Corresponding author(s): Juan Bolanos (jbolanos@usal.es)

Review Timeline:

Submission Date:	13th Oct 25
Editorial Decision:	4th Nov 25
Revision Received:	9th Nov 25
Accepted:	12th Dec 25

Editor: Esther Schnapp

Transaction Report:

Dear Juan,

Thank you for the submission of your Review to EMBO reports. We have now received the full set of referee reports that is pasted below.

As you will see, all referees acknowledge that the review is interesting and timely, which is great. All referees also have several suggestions for how the review could be improved and I think all suggestions are good and should be addressed, except may be point 3 by referee 3, this is up to you. I do think that either enriching the current figure, or better adding a second figure is a very good suggestion and will enhance the review. If the figures are generated with BioRender (or else) this needs to be mentioned in the figure legends.

As for timing, would it be possible for you to submit the revised version of the review by the end of November? Please let me know if this works for you. We will most likely not be able to publish it online in December but January should certainly work.

Thank you again for this nice contribution !

With best wishes,

Esther

Referee #1:

This review by Juan Bolaños and Ángeles Almeida is a strong and timely contribution that provides a clear and engaging synthesis on how astrocytic fatty acid oxidation functions as a signaling hub integrating ROS production, ketogenesis, and metabolic communication rather than serving merely as an energy source. The narrative and figures are coherent and well-structured. I have only minor editorial suggestions, which the authors may consider at their discretion.

- 1.If the authors agree, please replace "ETF-QOR" with the standard ETF-QO or the full protein name electron-transferring-flavoprotein dehydrogenase (ETFDH) as listed in UniProt. The figure could also include this protein to visually link it with the text.
- 2.Please ensure consistency in gene nomenclature: mouse genes should appear in lowercase italics (e.g., *Cpt1a*), and human genes in uppercase italics (e.g., *CPT1A*). For example, in the sentence: "In particular, astrocytes express higher levels of key FAO genes (e.g. *CPT1A*, *Acadl*, *Mtpα*) relative to neurons...".
- 3.Several references display minor formatting issues (duplicate years or stray digits after the year). The authors may wish to check their reference manager settings to correct these.
- 4.Since the review also discusses oligodendrocytes and β -oxidation, the authors could consider citing Asadollahi et al., 2024 (PMID: 39251890), which is directly relevant.
- 5.The paragraph on myelin changes following marathon running might benefit from a slightly more neutral phrasing, as the study by Ramos-Cabrer et al. did not specifically address β -oxidation or lipid remodeling nor did it establish a causal link to "cognitive resilience." The authors could clarify better that their interpretation is speculative, this would make the discussion even clearer for readers, especially to make some room to what still is unresolved and could be an interesting avenue for future research.

Referee #2:

This is an excellent review presenting clearly new mechanisms of action and roles of astrocytes in the context of brain function. It is very well written in an authoritative manner by leading figures in the field, Bolaños and Almeida, and includes up-to-date literature.

I only have a few minor comments for the authors' consideration which may help improve what is already an excellent piece. Broadly, my suggestions ask for a touch more context and specificity so non-expert readers can fully appreciate the significance of the advances, and for a clearer delineation of what is uniquely astrocytic versus shared with neurons. A brief integrative figure would also help synthesize the key concepts at a glance.

1. The title only reflects one aspect, even if a core one, of the review. Consider expanding it to encompass astrocytic signaling more globally (e.g., lipid signaling in astrocytes) or to better match the breadth of the content (e.g., astrocytic lipid metabolism as a central regulator of neuronal function via redox signaling and ketogenesis).
2. Page 3 "In astrocytes, this preferential FADH₂ input": please indicate more clearly "preferential" over what, so it is evident to readers not acquainted with bioenergetic mechanisms. A bit more background would help the non-expert understand what is

special here.

3. "In contrast to the conventional view of ROS as harmful byproducts, astrocytic ROS act as second messengers, modulating redox sensitive pathways and neuron-glia communication. ... in brain function." This section would benefit from a brief note on top papers in which ROS also act as signaling molecules in neurons (e.g., indicated in papers such as PMID: 29323696 or PMID: 38483244, though not necessarily these ones), so readers do not come away with the impression that such ROS functions are exclusive to astrocytes. Perhaps the first part of the sentence ("In contrast to the conventional view of ROS as harmful byproducts..") could be carefully qualified in this direction (as nicely done in the following paragraph for FAO in neurons).
4. "During fasting or ketogenic diets, peripheral ketone levels rise, but astrocytes may supplement this with locally produced ketones, maintaining CNS ketone availability." It would help to indicate, for the non-expert reader, the key brain functions ketones support (e.g., energetic support for synapses, redox balance, signaling roles) and why additional astrocyte-derived ketones might be important.
5. Page 6 "Thus, disruption of this ROS-mediated communication, by genetic inhibition of FAO impairs memory formation and synaptic plasticity (Morant-Ferrando et al., 2023; Vicente-Gutierrez et al., 2019)." It would be useful to add the animal model and the specific form(s) of synaptic plasticity reported, particularly as the next sentence(s) deal with *Drosophila*.
6. Page 7, last paragraph: it is not very clear which of the discussed actions are mediated directly by astrocytes, either intracellularly or via cell-cell communication. Please clarify these distinctions a bit more parsimoniously.
7. A figure integrating the key components of the review and the main insights would substantially increase its impact and ease of understanding.

Referee #3:

The review presented by Bolanos and Almeida discusses the different roles of astrocytic β -oxidation that have emerged in recent years. The manuscript is an easy-to-read overview and gives a nice summary of our knowledge about the importance of glial β -oxidation. In general, I strongly recommend publication, but I have some suggestions that the authors may want to consider.

1. I am surprised that the paragraph on lipid metabolism in oligodendrocytes does not cite the work of Asadollahi et al, 2024.
2. When discussing the role of astrocytic β -oxidation in allowing ROS-signaling that is important for neuronal function, e.g. in memory formation, the authors stress the evolutionary conservation of the function from flies to mammals. However, in other parts of the article this is not the case, even though also other functions of glial β -oxidation are conserved. For example, astrocytic ketogenesis has been shown to be essential for maintaining neuronal function, organismic homeostasis and animal survival under certain conditions in *Drosophila* (Schulz et al, 2015; Silva et al, 2022; McMullen et al, 2023). I suggest mentioning the evolutionary conservation also here.
3. Another role that has been proposed for glial LDs and β -oxidation is the detoxification of (peroxidated) lipids of neuronal origin (e.g. Ioannou et al, 2019; Liu et al, 2017). This role has not been discussed. I understand that it is conceptually somewhat different, since in this case the benefit does not come from a product of β -oxidation, but from removing the educts. Nonetheless, it might be worth mentioning.
4. The manuscript discusses the different roles of glial lipid metabolism, however, the figure focuses only on the production of ROS, instead of giving an overview over the roles. I would suggest to modify the figure to illustrate the different important effects of lipid metabolism (ROS, Ketone bodies, other signaling molecules, ...) to give the reader an easy overview. The figure should also be referenced in the text.

References:

- Asadollahi E, Trevisiol A, Saab AS, Looser ZJ, Dibaj P, Ebrahimi R, Kusch K, Ruhwedel T, Möbius W, Jahn O, et al (2024) Oligodendroglial fatty acid metabolism as a central nervous system energy reserve. *Nat Neurosci*
- Ioannou MS, Jackson J, Sheu S-H, Chang C-L, Weigel A V, Liu H, Pasolli HA, Xu CS, Pang S, Matthies D, et al (2019) Neuron-Astrocyte Metabolic Coupling Protects against Activity-Induced Fatty Acid Toxicity. *Cell* 177: 1522-1535.e14
- Liu L, MacKenzie KR, Putluri N, Maletić-Savatić M & Bellen HJ (2017) The Glia-Neuron Lactate Shuttle and Elevated ROS Promote Lipid Synthesis in Neurons and Lipid Droplet Accumulation in Glia via APOE/D. *Cell Metab*
- McMullen E, Hertenstein H, Strassburger K, Deharde L, Brankatschk M & Schirmeier S (2023) Glycolytically impaired *Drosophila* glial cells fuel neural metabolism via β -oxidation. *Nat Commun* 14: 2996
- Schulz JG, Laranjeira A, Van Huffel L, Gärtner A, Vilain S, Bastianen J, Van Veldhoven PP & Dotti CG (2015) Glial β -oxidation regulates *Drosophila* energy metabolism. *Sci Rep* 5: 7805
- Silva B, Mantha OL, Schor J, Pascual A, Plaçais P-Y, Pavlowsky A & Preat T (2022) Glia fuel neurons with locally synthesized ketone bodies to sustain memory under starvation. *Nat Metab* 4: 213-224

Referee #1

This review by Juan Bolaños and Ángeles Almeida is a strong and timely contribution that provides a clear and engaging synthesis on how astrocytic fatty acid oxidation functions as a signaling hub integrating ROS production, ketogenesis, and metabolic communication rather than serving merely as an energy source. The narrative and figures are coherent and well-structured. I have only minor editorial suggestions, which the authors may consider at their discretion.

1. If the authors agree, please replace "ETF-QOR" with the standard ETF-QO or the full protein name electron-transferring-flavoprotein dehydrogenase (ETFDH) as listed in UniProt. The figure could also include this protein to visually link it with the text.

Authors' reply: We thank the reviewer for this suggestion. We have now replaced ETF-QOR by ETFDH, and this protein is now included in the figure.

*2. Please ensure consistency in gene nomenclature: mouse genes should appear in lowercase italics (e.g., *Cpt1a*), and human genes in uppercase italics (e.g., *CPT1A*). For example, in the sentence: "In particular, astrocytes express higher levels of key FAO genes (e.g. *CPT1A*, *Acadl*, *Mtpα*) relative to neurons...".*

Authors' reply: Thanks for noting. This is now fixed.

3. Several references display minor formatting issues (duplicate years or stray digits after the year). The authors may wish to check their reference manager settings to correct these.

Authors' reply: Thanks for noting. This is now fixed.

4. Since the review also discusses oligodendrocytes and β -oxidation, the authors could consider citing Asadollahi et al., 2024 (PMID: 39251890), which is directly relevant.

Authors' reply: Thanks for noting and we apologize for the inadvertently omitted citation, which has now been included.

5. The paragraph on myelin changes following marathon running might benefit from a slightly more neutral phrasing, as the study by Ramos-Cabrer et al. did not specifically address β -oxidation or lipid remodeling nor did it establish a causal link to "cognitive resilience." The authors could clarify better that their interpretation is speculative, this would make the discussion even clearer for readers, especially to make some room to what still is unresolved and could be an interesting avenue for future research.

Authors' reply: We agree with the reviewer and, accordingly, this paragraph has now been amended following the reviewer' suggestion. (Lines 303-309).

Referee #2

This is an excellent review presenting clearly new mechanisms of action and roles of astrocytes in the context of brain function. It is very well written in an authoritative manner by leading figures in the field, Bolaños and Almeida, and includes up-to-date literature.

I only have a few minor comments for the authors' consideration which may help improve what is already an excellent piece. Broadly, my suggestions ask for a touch more context and specificity so non-expert readers can fully appreciate the significance of the advances, and for a clearer delineation of what is uniquely astrocytic versus shared with neurons. A brief integrative figure would also help synthesize the key concepts at a glance.

Authors' reply: We acknowledge the reviewer's positive comments on our review. We fully agree that more context would help non-experts to appreciate the significance of the advances. However, on the other hand, we agreed with the editor to propose a *mini-review* rather than a comprehensive full review, an excellent example of which was recently published and therefore available for the non-expert readership (please, see Vanherle et al., 2025). Following the suggestion of the reviewer, we have now generated a new Figure to enrich the key concepts and hence facilitate non-expert appreciation.

1. The title only reflects one aspect, even if a core one, of the review. Consider expanding it to encompass astrocytic signaling more globally (e.g., lipid signaling in astrocytes) or to better match the breadth of the content (e.g., astrocytic lipid metabolism as a central regulator of neuronal function via redox signaling and ketogenesis).

Authors' reply: We thank the reviewer for his/her suggestions for the title. We have considered several options based on the reviewer's comment and we believe that "Signaling roles of astrocytic fatty acid β -oxidation" is more direct otherwise keeping simplicity and catchy. We preferred to focus on the signaling role of β -oxidation, rather than in the supporting role of astrocytes for neurons -a topic recently and abundantly covered in several reviews.

2. Page 3 "In astrocytes, this preferential FADH₂ input": please indicate more clearly "preferential" over what, so it is evident to readers not acquainted with bioenergetic mechanisms. A bit more background would help the non-expert understand what is special here.

Authors' reply: We fully agree with the reviewer. Accordingly, we have now expanded this paragraph, and we hope it is now much clearer for a non-expert researcher in the field. (Lines 77-81).

3. "In contrast to the conventional view of ROS as harmful byproducts, astrocytic ROS act as second messengers, modulating redox sensitive pathways and neuron-glia communication. ... in brain function." This section would benefit from a brief note on top papers in which ROS also act as signaling molecules in neurons (e.g., indicated in papers such as PMID: 29323696 or PMID: 38483244, though not necessarily these ones), so readers

do not come away with the impression that such ROS functions are exclusive to astrocytes. Perhaps the first part of the sentence ("In contrast to the conventional view of ROS as harmful by products..") could be carefully qualified in this direction (as nicely done in the following paragraph for FAO in neurons).

Authors' reply: We fully agree with the reviewer and, accordingly, we have expanded this physiological view of ROS in other contexts, particularly in neurons. The paragraph was therefore amended and those two articles, which are very relevant, cited. (Lines 158-164).

4. "During fasting or ketogenic diets, peripheral ketone levels rise, but astrocytes may supplement this with locally produced ketones, maintaining CNS ketone availability." It would help to indicate, for the non-expert reader, the key brain functions ketones support (e.g., energetic support for synapses, redox balance, signaling roles) and why additional astrocyte-derived ketones might be important.

Authors' reply: We acknowledge the reviewer's comment on this issue. We have expanded the role for ketones in the brain and how is it thought they improve brain function, leaving open other still-to-decipher possibilities. (Lines 264-266).

5. Page 6 "Thus, disruption of this ROS-mediated communication, by genetic inhibition of FAO impairs memory formation and synaptic plasticity (Morant-Ferrando et al., 2023; Vicente-Gutierrez et al., 2019)." It would be useful to add the animal model and the specific form(s) of synaptic plasticity reported, particularly as the next sentence(s) deal with *Drosophila*.

Authors' reply: We thank the reviewer for this important comment. We have specified this paragraph, and we hope it is now much clearer. (Lines 174-178).

6. Page 7, last paragraph: it is not very clear which of the discussed actions are mediated directly by astrocytes, either intracellularly or via cell-cell communication. Please clarify these distinctions a bit more parsimoniously.

Authors' reply: We thank the reviewer for his/her suggestion. Accordingly, we have now expanded and clarified this paragraph. (Lines 226-239).

7. A figure integrating the key components of the review and the main insights would substantially increase its impact and ease of understanding.

Authors' reply: We thank the reviewer for his(her suggestion. We have now added one Figure to the manuscript, and we hope that it will increase the ease in understanding.

Referee #3:

The review presented by Bolanos and Almeida discusses the different roles of astrocytic β -oxidation that have emerged in recent years. The manuscript is an easy-to-read overview and gives a nice summary of our knowledge about the importance of glial β -oxidation. In general, I strongly recommend publication, but I have some suggestions that the authors may want to consider.

1. I am surprised that the paragraph on lipid metabolism in oligodendrocytes does not cite the work of Asadollahi et al, 2024.

Authors' reply: We thank the reviewer for his/her helpful comment. We apologize for the inadvertently omitted citation, which has now been included.

*2. When discussing the role of astrocytic β -oxidation in allowing ROS-signaling that is important for neuronal function, e.g. in memory formation, the authors stress the evolutionary conservation of the function from flies to mammals. However, in other parts of the article this is not the case, even though also other functions of glial β -oxidation are conserved. For example, astrocytic ketogenesis has been shown to be essential for maintaining neuronal function, organismic homeostasis and animal survival under certain conditions in *Drosophila* (Schulz et al, 2015; Silva et al, 2022; McMullen et al, 2023). I suggest mentioning the evolutionary conservation also here.*

Authors' reply: We agree with the reviewer' suggestion and this has now been fixed. (Lines 120-121).

3. Another role that has been proposed for glial LDs and β -oxidation is the detoxification of (peroxidated) lipids of neuronal origin (e.g. Ioannou et al, 2019; Liu et al, 2017). This role has not been discussed. I understand that it is conceptionally somewhat different, since in this case the benefit does not come from a product of β -oxidation, but from removing the educts. Nonetheless, it might be worth mentioning.

Authors' reply: We agree with the reviewer' suggestion and we have now mentioned this aspect, citing the work by Liu et al (2017) in addition to that by Ioannou et al. (2019), which was already cited in the original version. (Lines 264-266).

4. The manuscript discusses the different roles of glial lipid metabolism, however, the figure focuses only on the production of ROS, instead of giving an overview over the roles. I would suggest to modify the figure to illustrate the different important effects of lipid metabolism (ROS, Ketone bodies, other signaling molecules ,...) to give the reader an easy overview. The figure should also be referenced in the text.

Authors' reply: We thank the reviewer for his/her suggestion. Accordingly, we have now enriched the figure and generated a new one to increase the the content and provide the reader an easy overview.

Prof. Juan Bolanos
Universidad de Salamanca
Institute of Functional Biology and Genomics
Zacarias Gonzalez 2
Salamanca 37007
Spain

Dear Juan,

I am pleased to inform you that your review has been accepted for publication in EMBO reports. Your manuscript will be processed for publication by EMBO Press. It will be copy edited and you will receive page proofs prior to publication.

When SpringerNature is asking you to sign the license agreement form please enter this token so that no publication charges will apply:

Token Unavailable

Best wishes,
Esther

>>> Please note that it is EMBO Reports policy for the transcript of the editorial process (containing referee reports and your response letter) to be published as an online supplement to each paper. If you do NOT want this, you will need to inform the Editorial Office via email immediately. More information is available here: <https://link.springer.com/partners/embo-press/editorial-policies#Peer%20review>